# Neural Effects of One’s Own Voice on Self-Talk for Emotion Regulation

**DOI:** 10.3390/brainsci14070637

**Published:** 2024-06-26

**Authors:** Hye-jeong Jo, Chanmi Park, Eunyoung Lee, Jee Hang Lee, Jinwoo Kim, Sujin Han, Joohan Kim, Eun Joo Kim, Eosu Kim, Jae-Jin Kim

**Affiliations:** 1Graduate School of Medical Science, Brain Korea 21 Project, Yonsei University College of Medicine, Seoul 03722, Republic of Korea; hjjo2233@yuhs.ac (H.-j.J.); kimeosu@yuhs.ac (E.K.); 2HCI Lab, Cognitive Science, Yonsei University, Seoul 03722, Republic of Korea; chanmipark@yonsei.ac.kr (C.P.); eunyounglee94@yonsei.ac.kr (E.L.); jinwoo@haii.io (J.K.); 3Department of Human-Centered Artificial Intelligence, Sangmyung University, Seoul 03016, Republic of Korea; jeehang@smu.ac.kr; 4Department of Communication, Yonsei University, Seoul 03722, Republic of Korea; h_sj0525@yonsei.ac.kr (S.H.); jkim@yonsei.ac.kr (J.K.); 5Graduate School of Education, Yonsei University, Seoul 03722, Republic of Korea; ctl-kej@yonsei.ac.kr; 6Department of Psychiatry and Institute of Behavioral Science in Medicine, Yonsei University College of Medicine, Seoul 03722, Republic of Korea

**Keywords:** one’s own voice, emotion regulation, self-affirmation, cognitive defusion, fMRI

## Abstract

One’s own voice undergoes unique processing that distinguishes it from others’ voices, and thus listening to it may have a special neural basis for self-talk as an emotion regulation strategy. This study aimed to elucidate how neural effects of one’s own voice differ from those of others’ voices on the implementation of emotion regulation strategies. Twenty-one healthy adults were scanned using fMRI while listening to sentences synthesized in their own or others’ voices for self-affirmation and cognitive defusion, which were based on mental commitments to strengthen one’s positive aspects and imagining metaphoric actions to shake off negative aspects, respectively. The interaction effect between voice identity and strategy was observed in the superior temporal sulcus, middle temporal gyrus, and parahippocampal cortex, and activity in these regions showed that the uniqueness of one’s own voice is reflected more strongly for cognitive defusion than for self-affirmation. This interaction was also seen in the precuneus, suggesting intertwining of self-referential processing and episodic memory retrieval in self-affirmation with one’s own voice. These results imply that unique effects of one’s own voice may be expressed differently due to the degree of engagement of neural sharpening-related regions and self-referential networks depending on the type of emotion regulation.

## 1. Introduction

Self-talk is the way people talk to themselves; it is their inner voice. A functional description of self-talk includes self-directed verbal expressions, encompasses various dimensions, such as positive or negative, overt or covert, and instructional or motivational, and involves elements of interpretation linked to the context of the statements used [1]. While inner speech is an unstructured stream of mental activity that includes voluntary or involuntary thought and reflection, self-talk is an attempt at self-regulation that occurs in response to or anticipation of a specific event or situation [2]. In particular, positive self-talk for self-affirming may play a crucial role in influencing decision-making [3], facilitating emotion regulation [4], and adapting to challenges [5], and thus has been employed in a variety of activities, such as enhancing performance in sports [6,7], boosting academic involvement [8], and managing anxiety in public speaking [9].

Self-talk inevitably leads to hearing one’s own voice. Individuals process their own voices differently from others’ voices in ways that lead them to perceive their own voice as more attractive [10,11]. This attractiveness can be explained by vocal implicit egoism, a form of self-enhancement driven by the familiarity effect and self-positivity bias [12]. Phonetic realizations of one’s own voice significantly shape phonological contrasts, leading to more accurate recognition of words in one’s own voice compared to the voices of others [13]. In addition, as individuals become accustomed to their own voices through lifelong exposure, hearing their own voice exhibits the phenomenon of neural sharpening, in which more common stimuli reduce neural responses to them, and thus it lowers the level of activation of the superior temporal gyrus (STG), which is involved in neural sharpening for voices [14]. Furthermore, a previous neuroimaging study have shown that hearing the own voice causes engagement of the self-referential network, including the medial prefrontal and parietal cortices [15], supporting that it is linked to self-awareness in speech processing. Since people listen to their own voices while speaking, they perceive their voices more deeply and richly through bone conduction and air conduction. Sometimes people listen to their own recorded voices, which they hear only through air conduction, making them feel uncomfortable because they are different from their familiar voices [16]. Therefore, if an experimental attempt is planned to investigate the effect of self-talk, it is preferable to listen to one’s own recorded voice rather than someone else’s voice, and the process of converting this recorded voice to sound like the voice heard when speaking is first required. This kind of investigation may be possible by measuring electrodermal activity, a physiological signal for objective assessment of emotional states [17], or functional MRI, a powerful technique that captures brain responses to task-related activities with high spatial resolution [18].

Self-talk may be one of emotion regulation strategies. Emotion regulation is a process of controlling one’s own emotional state. A variety of emotion regulation strategies have been developed to improve mental health in several different categories, such as attention allocation, response regulation, reappraisal, and suppression [19,20]. These strategies have been reported to be associated with multiple cortical and subcortical activations in the brain [21,22]. Self-talk for self-affirming may be an example of practical regulatory attempts. Self-affirmation refers to the process of focusing on and acknowledging one’s positive qualities, values, and strengths such as “I am confident.” or “I am getting better and better every day.” Self-affirmation theory suggests that individuals protect their self-integrity by affirming unrelated values when faced with threats that make them feel like they do not meet cultural or social standards, and they are less inclined to distort or overly interpret the threat and can respond to the situation more impartially and openly [23]. Because of these characteristics, self-affirmation helps people manage their emotions and lessen negative emotions, as exemplified by improving self-integrity [24] and reducing dissonance-related discomfort [25]. Moreover, self-affirmation may exert a stress resilience effect by influencing physiological responses, such as decreasing cortisol reactivity in stressful situations [26] and lowering epinephrine levels during exams [27]. Supporting this, self-affirmation increases activity in self-referential processing-related regions, such as the medial prefrontal cortex (MPFC) and posterior cingulate cortex [28,29] as well as reward processing-related regions, such as the ventral striatum [28,30].

However, self-talk for self-affirming is not effective for everyone. For example, individuals with lower self-esteem may experience more negative emotions when employing positive self-talk [31]. Our research group found that after exposure to self-affirming stimuli, individuals with low life satisfaction displayed increased functional connectivity in the MPFC compared to those with high life satisfaction [32], suggesting that the effects of self-talk for self-affirming may vary depending on individuals’ ongoing emotional state. Therefore, other methods may be needed to induce positive behavioral changes in individuals for whom the effect of self-talk for self-affirming is weakened, and mindfulness and acceptance-based concepts can be considered promising alternatives [33].

Mindfulness and acceptance emphasize cultivating present-moment awareness and non-judgmental observation of one’s thoughts, feelings, and physical sensations [34]. In this approach, there is no need to change or reduce emotions or thoughts, regardless of their nature or valence. Acceptance and commitment therapy (ACT) is a form of psychotherapy that integrates mindfulness and acceptance strategies with behavior change approaches, and increases one’s psychological flexibility, the ability to adapt behavior to the ongoing experiences of life, and to make choices based on what is truly meaningful to an individual [35]. Like self-affirming self-talk, mindfulness and acceptance reduce stress [36], improve attention and concentration [37], effectively regulate emotion [38] and increase resilience [39], and thus have been used in sport performance [40], meditation [41], and clinical settings [42]. Neuroimaging studies related to ACT have also been conducted. In a study for individuals with obsessive compulsive disorder, for example, it led to an increased activity in various brain regions, such as the inferior frontal gyrus, superior temporal gyri, posterior cingulate cortex (PCC), precuneus, and insula, suggesting that the therapeutic benefits of ACT may be linked to complex processes related to salience, interoception, multisensory integration, language, and self-reflection [43].

There are six core processes in ACT for cultivating psychological flexibility, called ‘hexaflex’, and one of them is cognitive defusion, the ability to step back from thoughts and view them as mental processes, which refers to a process aimed at changing our relationship with our cognitive and verbal processes [44]. This empowers individuals to engage in behaviors despite distressing thoughts and emotions, enabling them to choose which experiences to hold onto and which to let go. There was a report that when a single-word repetition exercise for cognitive defusion was applied to negatively evaluated self-referential content, it induced a significant reduction in believability and discomfort associated with negative self-relevant words compared to thought controls such as positive self-talk [45]. A similar intervention showed decreased discomfort and increased willingness and believability of negative statements presented relative to non-defused statements [46].

When individuals become fused with their thoughts, they become entangled in their thinking patterns, believing that their thoughts are an accurate representation of reality. In ACT, cognitive defusion is highlighted for creating distance between individuals and their thoughts, and this defuses from their negative content makes individual to observe their thinking processes from a more objective standpoint [44]. Another approach to cognitive defusion involves encouraging individuals to pay attention to their thoughts by using metaphoric phrases [44,47]. For instance, they might visualize their thoughts as leaves gently falling on a river (leaves on a stream) or as posters being carried by soldiers in a parade (soldiers in the parade). A previous neuroimaging study reported that cognitive defusion decreased neural activity in subcortical areas, such as the brainstem, thalamus, hippocampus, and left amygdala [48], suggesting that approaching negative stimuli with emotional detachment may result in a reduction in emotional intensity and physiological stress responses.

The current study used the task of listening to sentences of the emotion regulation strategies and assessing their emotional influence while undergoing functional MRI. Given the uniqueness of one’s own voice, it may be more efficient to perform this task by listening to one’s own voice rather than listening to the voices of others, and this efficiency may vary in degree depending on the type of emotion regulation strategy. The purpose of the current study was to elucidate how the neural effects of one’s own voice differ from those of others’ voices on implementing the voice-listening emotion regulation strategies, such as self-affirmation and cognitive defusion. For this purpose, functional MRI scans were taken while performing a task of listening to sentences of the two strategies synthesized in one’s own voice or another’s voice before the experiment. To our knowledge, an fMRI study like this has not been conducted before and will contribute to providing a foundation and understanding of the importance of using one’s own voice in the development of emotion regulation strategies. Based on the previous findings on the neural effects of hearing the own voice, we hypothesized that the engagement of neural sharpening-related regions and self-referential networks by listening to one’s own voice would differ depending on the type of emotion regulation strategy.

## 2. Materials and Methods

### 2.1. Participants

A total of 27 healthy right-handed adults (12 males/15 females) were initially recruited into the study. Sample size was calculated through a power analysis (G*Power 3) [49] with a medium-sized effect of 0.5 and 80% power at an alpha level of 0.05. Exclusion criteria included the presence of a neurological, psychiatric, or significant medical illness, current or past history of substance abuse or dependence, and participation in a psychotherapeutic setting. Due to technical difficulties in synthesizing one’s own voice during the process of generating audio–visual stimuli for fMRI scanning, the stimuli for six participants did not reach a level suitable for the experiment. Accordingly, the remaining 21 participants (8 males/13 females, mean age ± standard deviation: 25.6 ± 4.3) for whom auditory–visual stimuli were successfully prepared participated in the fMRI experiment.

This study was approved by the Institutional Review Board of Severance Hospital, Yonsei University, South Korea (4-2018-0853), and carried out in accordance with the Declaration of Helsinki. All participants gave informed consent after receiving sufficient explanations about the purpose and process of the study from the researcher prior to the study participation.

### 2.2. Psychological Assessments

All participants were evaluated for psychological factors that could affect positive self and self-criticism using the two self-report questionnaires. They were the Rosenberg Self-Esteem Scale (RSES), consisting of 10 items with a 4-point Likert scale from 1 (strongly disagree) to 4 (strongly agree) and a total score that ranges from 10 to 40 [50], and the Levels of Self-Criticism Scale (LOSC), consisting of a 22-item with a 7-point Likert scale anchored by 1 (not at all) and 7 (very well) [51]. The LOSC includes two subscales: comparative self-criticism (12 items, total score range: 12–82) and internalized self-criticism (10 items, total score range: 10–70).

### 2.3. Audiovisual Stimuli and Experimental Procedure

Participants performed an emotion regulation task during MRI scans. The task was to listen to a short sentence with one of three different contents, such as self-affirmation, cognitive defusion, and neutral, in their own voice or someone else’s voice, and evaluate how emotionally the sentence affected them. The sentences of self-affirmation and cognitive defusion for the experimental conditions (Figure 1A) were provided in the form of metaphorical distancing for their self-respect (e.g., I believe that I can always achieve my dream) and distress (e.g., I scatter the tears of hurt in the calm waves), respectively. The neutral sentences for the control condition were generated in the form of simple event descriptions (e.g., I can pay by credit card or cash). Each of the three different contents consisted of 20 sentences, and thus a total of 60 sentences were prepared (see Appendix A). Since all these sentences were reproduced with the participant’s own voice and the voice of another person, a total of 120 voice stimuli were used in the task. In the actual experiment, the sentences were expressed in the form of audio playback along with visual presentation on a screen of black letters on a white background. Presentation of voice stimuli was conducted through MR-compatible headphones that delivered sound stimuli and attenuated ambient noise of the scanner.

A schematic diagram of the overall experimental procedure is presented in Figure 1B. In order to assess the emotional impact of first hearing these sentences during fMRI task performance without being familiar with them, vocal stimuli were generated as synthesized voices rather than reading and recording the sentences. Tacotron, which can be learned with text and audio pairs, was used for synthesis, and the underlying generation model was an end-to-end Text-to-Speech that could receive text as input and generate a raw spectrogram immediately [52]. Tacotron was trained using only text and audio pairs collected from all participants’ recorded voices. First, data design was needed as a preliminary preparation for voice synthesis. Then, 10 learning sentences were prepared for each of the 60 sentences to be used in the fMRI experiment, and a total of 600 such sentences consisted of sentences with similar pronunciation and domain-specific word sentences. Specifically, the sentences were constructed by combining specific sets of syllables with their corresponding sound units (phonemes), and each of these syllables comprised a sequence of three phonemes (triphones) [53]. Next, one week before the fMRI experiment, the participants visited the laboratory and recorded all learning sentences for about an hour. Before voice synthesis was performed on the collected learning data, a tone control step was performed for each data to reduce the heterogeneity of the recorded voice. Based on the voice frequency of 1000 Hz, the high frequency part was lowered by 2 dB, and the low frequency part was emphasized by 2 dB [54,55]. After completing the data preprocessing process, the final 60 vocal stimuli were derived by recording the voice file using Tacotron. The voice of another person was synthesized in the same way before the experiment with a voice of the same sex and similar age unfamiliar to the participants.

In the sequence of the fMRI experiment, 120 pre-generated audio–visual stimuli were randomly placed, and each stimulus was presented for seven seconds at jittered intervals of one to seven seconds (Figure 1C). Two sets of these randomly arranged stimuli were produced and presented alternately to the participants. Given the length of time required for the task, the sequence was split into two sessions, each consisting of 60 trials, totaling 9 min. In addition, the visual stimuli included a voice-matched sentence, plus the question “How much does this sentence affect your emotions?” and a choice of “not at all”, “somewhat”, “moderately”, and “strongly”, and participants were instructed to press one button corresponding to each of the four options. During the 7 s trial, the actual voice stimuli were up to 4 s in length, allowing participants to have at least 3 seconds to answer the question.

### 2.4. Imaging Data Acquisition and Preprocessing

MRI data were acquired on a 3-Tesla scanner (Magnetom Verio; Siemens Medical Solutions, Erlangen, Germany). Functional images were collected using an echo planar sequence (echo time = 30 ms; repetition time = 2000 ms; flip angle = 90; slice thickness = 3 mm; field of view = 240 mm; and matrix = 64 × 64). T1-weighted images were also collected using a 3D spoiled gradient-recall sequence (echo time = 2.46 ms; repetition time = 1900 ms; flip angle = 9; slice thickness = 1 mm; number of slices = 176; and matrix size = 256 × 256). Using the Statistical Parametric Mapping 12 (Welcome Department of Cognitive Neurology, Institute of Neurology, London, UK) and MATLAB 2020a (Mathworks, Natick, MA, USA), the following image-preprocessing steps were conducted in order: realignment on the first image, slice-timing correction, co-registration and spatial normalization using a standard Montreal Neurological Institute template, and smoothing using a Gaussian kernel with a full width at half maximum of 8 mm.

### 2.5. Behavioral Response Analysis

Behavioral responses during task performance were recorded to calculate an emotional influence score, which was scored from 0 for “not at all” to 3 for “strongly”, suggesting that the higher the score, the more positively the stimulus affected the participant’s emotions. In order to assess the main effect of each of two categorical independent variables, voice identity and emotion regulation strategy, and the interaction effect between them, the emotional influence scores were compared in a 2 (identity: own and the other) × 3 (strategy: self-affirmation, cognitive defusion, and neutral) manner using two-way analysis of variance (ANOVA). In addition, post hoc Bonferroni t-tests were used to determine the significant pairs.

### 2.6. Imaging Data Analysis

Preprocessed functional data were analyzed using a general linear model at the single-subject level. Experimental trials were modeled separately using a canonical hemodynamic response function for individual data. Multiple linear regression was used to obtain parameter estimates using a least-squares approach. These estimates were further analyzed by testing specific contrasts using the participant as a random factor. Contrast images of four experimental conditions subtracted by the neutral control condition were created for each participant on the first-level analysis. Individual realignment parameters were entered as regressors to control for movement-related variance. In order to find brain activations in each experimental condition, the contrast images were entered into the one-sample *t*-test and the full factorial model across the participants. In addition, in order to find common activation areas of the two emotion regulation strategies, a conjunction analysis was performed between contrast images of the self-affirmation and cognitive defusion conditions.

Second-level analysis was executed in a 2 (identity) × 2 (strategy) repeated-measures ANOVA on contrast images obtained in the first-level analysis to identify brain regions showing the main and interaction effects. For all above image analyses, statistical threshold was set at voxel-level *p* < 0.001 (uncorrected) at first, and then all clusters that the met false discovery rate (FDR)-corrected *p* < 0.05 at the cluster level were considered significant. Post hoc analysis was performed to identify the direction of differences by extracting parameter estimates of each significant cluster with a radius of 5 mm sphere using MarsBaR 0.44. Additionally, the regional activity values were used to calculate the correlations with psychological assessment scores, such as the RSES score and two subscale scores of the LOSC. Considering the correlations with the three scores, the significance level in this analysis was set at *p* = 0.016 (0.05/3).

## 3. Results

### 3.1. Behavioral Data

The mean emotional influence score was 1.64 ± 0.79 in the own-voice self-affirmation condition, 1.07 ± 0.71 in the own-voice cognitive defusion condition, 0.75 ± 0.85 in the own-voice neutral condition, 1.59 ± 0.78 in the other-voice self-affirmation condition, 1.06 ± 0.59 in the other-voice cognitive defusion condition, and 0.72 ± 0.78 in the other-voice neutral condition (Figure 2). Two-way ANOVA for 2 (voice identity: own and the other) × 3 (emotion regulation strategy: self-affirmation, cognitive defusion, and neutral) showed no main effect of identity (*F*_1, 41_ = 0.04, *p* = 0.84), but main effect of strategy was significant (*F*_2, 61_ = 14.72, *p* < 0.001). Post hoc tests showed that the emotional influence scores were significantly higher in the self-affirmation strategy than in the cognitive defusion strategy (*t*_20_ = 3.37, *p* = 0.003) and neutral content (*t*_20_ = 5.37, *p* <0.001), and did not significantly differ between the cognitive defusion strategy and neutral content. In addition, there was no interaction effect between identity and strategy (*F*_2, 124_ = 0.01, *p* = 0.99).

### 3.2. Activation Related to Each Experimental Condition

Table 1 presents the increased activity during four different experimental conditions compared to the neutral condition. There was no significant result in the own-voice self-affirmation condition, but the own-voice cognitive defusion condition showed activation in various cortical and subcortical regions. The other-voice self-affirmation condition showed no significant result, but the other-voice cognitive defusion condition showed activation of the left lingual gyrus. Meanwhile, there was no significant region showing common activation between self-affirmation and cognitive defusion in the conjunction analysis.

### 3.3. Neural Effects of Voice Identity and Emotion Regulation Strategy

Table 2 presents the neural regions showing the main effects of identity and strategy and their interaction effects in the 2 (voice identity: own and the other) × 2 (emotion regulation strategy: self-affirmation and cognitive defusion) repeated-measures ANOVA. There was no main effect of identity, but the main effect of strategy was identified in the bilateral MPFC, left premotor cortex, left STG, left inferior temporal gyrus, and right fusiform gyrus. Post hoc analysis indicated that among these regions, the bilateral MPFC and left STG responded more to the self-affirmation condition than to the cognitive defusion condition, whereas the left premotor cortex, left inferior temporal gyrus, and right fusiform gyrus responded more to the cognitive defusion condition than to the self-affirmation condition.

The interaction effect between identity and strategy was found in multiple regions, including the right superior temporal sulcus (STS), right middle temporal gyrus, bilateral parahippocampal cortex, bilateral precuneus, and bilateral calcarine cortex. The differences among the conditions in these regions are presented in Figure 3. The left parahippocampal cortex showed that cognitive defusion led to higher activity by own voice than by the other voice (*t*_20_ = 1.94, *p* < 0.05), whereas the right parahippocampal cortex displayed that own voice led to higher activity in the cognitive defusion condition than in the self-affirmation condition (*t*_20_ = 2.36, *p* < 0.05). The right STS and right middle temporal gyrus also showed that own voice led to higher activity in the cognitive defusion condition than in the self-affirmation condition (*t*_20_ = 2.76, *p* < 0.05; *t*_20_ = 2.21, *p* < 0.01, respectively). In addition, the other voice in the cognitive defusion condition induced higher activity in the bilateral precuneus than own voice in the self-affirmation condition (*t*_20_ = 3.09, *p* < 0.01) and in the bilateral calcarine cortex than the other three conditions (*p* < 0.001 in all comparisons).

### 3.4. Correlations between Neural Activity and Psychological Assessments

Among the brain regions showing the interaction effect between identity and strategy, significant correlations were observed only in bilateral precuneus and right STS activity in the own-voice conditions as follows (Figure 4). Bilateral precuneus activity in the own-voice self-affirmation condition and own-voice cognitive defusion condition showed a negative correlation with the RSES scores at a significant level and at a trend level, respectively (*r* = −0.53, *p* = 0.013; *r* = −0.50, *p* = 0.021, respectively) and a positive correlation with the LOSC–internalized self-criticism scores at a significant level (*r* = 0.70, *p* < 0.001; *r* = 0.60, *p* < 0.01, respectively). Similarly, right STS activity in the own-voice self-affirmation condition was negatively correlated with the RSES scores at a trend level (*r* = 0.47, *p* = 0.033), and activity in the own-voice self-affirmation condition and own-voice cognitive defusion condition showed a positive correlation with the LOSC–internalized self-criticism scores at a significant level and at a trend level, respectively (*r* = 0.63, *p* < 0.01; *r* = 0.47, *p* = 0.034, respectively).

## 4. Discussion

The current study was performed to reveal differences in the neural effects between one’s own and others’ voices on implementing the voice-listening emotion regulation strategies. Since there was no main effect of voice identity and no interaction effect of voice identity and emotion regulation strategy in the behavioral results, it can be stated that a difference in voice identity did not lead to a significant difference in the assessment of emotional influence. However, even so, these results do not rule out the possibility that a cognitive difference in voice identity may cause a difference in the neuroimaging results.

Notably, neuroimaging analysis did not show the main effect of voice identity, but revealed the interaction effects between voice identity and emotion regulation strategy. These meaningful results may be due to the impact of a cognitive difference in voice identity. Post hoc test showed higher activity in three right temporal lobe regions, including the STS, middle temporal gyrus, and parahippocampal cortex, during the own-voice cognitive defusion condition compared to the own-voice self-affirmation condition, suggesting that the uniqueness of one’s own voice is reflected more strongly in the functional involvement of these regions for cognitive defusion than for self-affirmation. The STS plays an important role in multiple functions of speech processing, audio–visual integration, theory of mind, and biological motion perception [56]. The middle temporal gyrus is also involved in audio–visual integration as a key element of the semantic system along with STS [57,58]. The parahippocampal cortex not only plays an important role in episodic memory encoding and retrieval but is also critical for visuospatial processing and processing of scene information [59,60]. Taken together, our results indicate that demands for the integration of audio–visual information and scene information in cognitive defusion may be processed more efficiently when presented in one’s own voice rather than in others’ voices. Furthermore, our post hoc test also showed that left parahippocampal cortex activity was significantly higher in the own-voice cognitive defusion condition than in the other-voice cognitive defusion condition, suggesting that engagement of episodic memory retrieval in cognitively distancing tasks may occur more clearly when using one’s own voice.

The second thing worth mentioning in the results of the interaction effect is that bilateral precuneus activity was significantly reduced in the own-voice self-affirmation condition compared to the other-voice cognitive defusion condition. The precuneus, along with the MPFC, is involved in self-referential processing as part of the default mode network [61,62]. This network is responsible for integrating moment-to-moment external information with prior information, and thus its activity is influenced by context and incoming input [63]. The precuneus also has several other functions, including episodic memory retrieval, which is closely intertwined with self-referential processing [64]. Therefore, our result in the precuneus may imply that self-affirmation with one’s own voice may be related to this close intertwinement. However, contrary to the previous research finding that self-affirming stimulation caused an increase in activity related to self-referential processing in the posterior cingulate cortex [28], which is close to the precuneus, our result showed a decrease in precuneus activity, which is not easy to explain. One possible explanation is that since episodic memory retrieval operates more strongly in cognitive defusion, which was the comparison condition, it appears to be relatively reduced in self-affirmation. Alternative explanation is that just as the familiarity of one’s own voice due to long-term exposure leads to the phenomenon of neural sharpening, which is associated with a decrease in STG activity [14], a similar mechanism may be at work for self-affirmation of one’s own voice in the precuneus.

Meanwhile, in correlation analysis, bilateral precuneus activities in the own-voice self-affirmation condition were negatively correlated with the RSES scores and positively with the LOSC–internalized self-criticism scores, and those in the own-voice cognitive defusion condition showed the similar results. These results may mean that people with lower self-esteem and more self-criticism engage in more self-referential processing or episodic memory retrieval while performing emotion regulation strategies, whether self-affirmation or cognitive defusion, in their own voice, and thus these people will be able to benefit from properly handing self-referential memory retrieval through such strategies. A recent study reported that the default mode network including the precuneus may be involved in external naturalistic event processing and prediction-based learning [65], suggesting that this network can be changed by applying an appropriate learning strategy. Previous literature has revealed that emotion regulation strategies, including mindfulness, are effective in controlling pathological factors, such as anxiety, depression, and stress, that underlie numerous clinical diseases [66,67]. Our findings may provide a brain basis for the effectiveness of using one’s own voice to implement such strategies.

Another characteristic result was that activity of the calcarine cortex was increased only in the other-voice cognitive defusion condition compared to the other three conditions. This region, the primary visual cortex, is the most important area engaged in visual mental imagery and is especially related to its vividness [68,69]. In the same way as the previous explanation that cognitive defusion may lead to visual association cortex activity as it causes visual mental imagery, it can also lead to increased activity in the primary visual cortex. However, our finding that this increase was observed only in the other-voice cognitive defusion condition suggests that the vividness of visual mental imagery can be more intense when the voices of others are presented rather than one’s own.

Meanwhile, interesting results also emerged from the differences between the two emotion regulation strategies. In the behavioral responses, participants reported positive responses to self-affirmation in comparison with cognitive defusion and neutral sentences. This may be because self-affirmation stimuli consist of sentences related to self-respect, which aided participants to make quick, intuitive judgments when choosing an emotional influence, unlike cognitive defusion stimuli consisting of metaphorical sentences. In addition, the main effect of strategy revealed the involvement of completely different brain regions between the two strategies, whereas the conjunction analysis to determine what brain regions were common between the two strategies yielded no meaningful results.

In other words, the MPFC activated more for self-affirmation, whereas the premotor cortex activated more for cognitive defusion. Since self-affirmation is based on self-related processing, increased MPFC activity during the implementation of this strategy results is consistent with previous results showing that self-referential brain regions are involved in self-affirmation [28,29]. Previous studies reported that self-affirmation also involved reward-related regions [28,30], which were not found in our strategy effect. This lack of significant results may be due to the possibility that cognitive defusion also acted as a reward, and thus the overlap effect cancelled each other out. Unlike self-affirmation, cognitive defusion involved imagining actions related to metaphorical contents, and thus it would activate brain regions that underlie this imagining function, which may have led to increased activity in the premotor area. This interpretation is consistent with by previous findings that the premotor cortex is involved in internal imagery, when participants imagine themselves performing specific actions [70,71].

Another interesting result from the strategy effect was that self-affirmation and cognitive defusion led to greater activation of the auditory association cortex, i.e., STG, and visual association cortex, i.e., inferior temporal gyrus and fusiform gyrus, respectively. The former result appears to be because the contents of self-affirmation in our task allow for more natural self-talk and stimulate more internal language processing than relatively unfamiliar cognitive defusion. On the other hand, the latter result seems to be due to stimulation of visual images using metaphoric distancing statements. In particular, it is interesting to note that the involvement of the fusiform gyrus was also observed in another study, which addressed cognitive defusion using a different paradigm that asked participants to process negative emotional pictures by distancing themselves from emotional events [48]. In our experiment, participants were required to imagine scenarios in which they performed actions, such as putting thoughts on the leaves and send them down to the river and putting haunting thoughts into the bubbles and pop them, to create a distance from emotional stressors. This visual mental imagery might lead to the involvement of the inferior temporal gyrus and fusiform gyrus, which have consistently been reported to be involved in visual mental imagery [72,73,74].

There are some limitations in the current study. First, due to the small sample size and the age of participants skewed towards their 20s, it is difficult to generalize the result. Second, in designing the experimental task, we tried to make it as similar to a self-talk situation as possible. However, considering that self-talk is actually speaking out loud and listening to it at the same time, it is difficult to say that this task, which uses listening to pre-synthesized speech, truly reflects self-talk. Third, our task may be effective for emotional regulation when performed in a quiet environment, but the strong noise in the MRI scanning situation may have interfered with this environment, and thus the full effect may not have been achieved. Additionally, the effects of these types of emotion regulation may appear differently depending on how familiar the participants are with them, but this familiarity was not assessed in our study.

## 5. Conclusions

In the current study for elucidating differences in the neural effects between one’s own and others’ voices on implementing the voice-listening emotion regulation strategies, there was no main effect of voice identity, but the interaction effects between voice identity and emotion regulation strategy were revealed. In particular, the STS, middle temporal gyrus, and parahippocampal cortex showed higher activity during the own-voice cognitive defusion condition compared to the own-voice self-affirmation condition, suggesting that the uniqueness of one’s own voice is reflected more strongly for cognitive defusion than for self-affirmation. In addition, we found meaningful results regarding bilateral precuneus activity implying that self-affirmation with one’s own voice may be associated with the intertwinement of self-referential processing and episodic memory retrieval, and calcarine cortex activity suggesting that the vividness of visual mental imagery for cognitive defusion can be more intense in the voices of others. These results suggest that, as stated in the hypothesis, the unique effects of one’s own voice may be expressed differently due to the degree of engagement of neural sharpening-related regions and self-referential networks depending on the type of emotion regulation strategy. These insights into brain responses may be important for developing personalized treatment approaches to improve mental health, taking into account individual differences and preferences. From this perspective, future research will be needed to determine how one’s own voice affects the brain in emotional regulation strategies other than self-affirmation and cognitive defusion.

## Figures and Tables

**Figure 1 brainsci-14-00637-f001:**
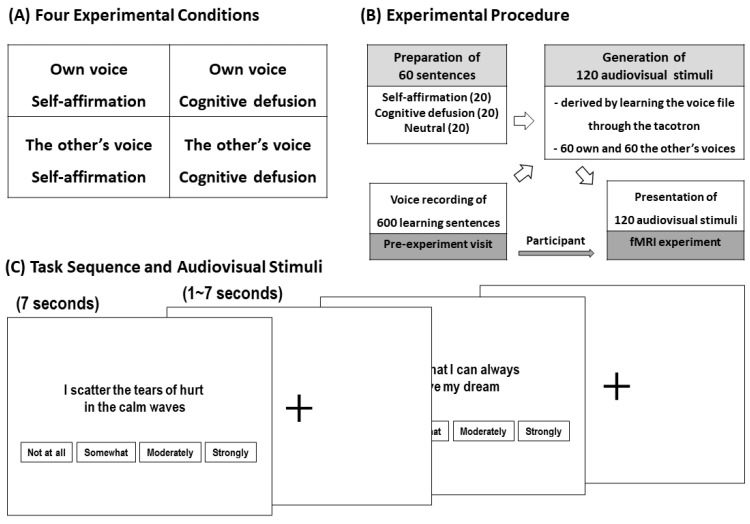
The experiment overview: (**A**) Four experimental conditions, such as the own-voice and self-affirmation, the own-voice and cognitive defusion, the other-voice and self-affirmation, and the others-voice and cognitive defusion conditions. (**B**) Schematic diagram of the experimental procedure, including participant visitation and preparation of experimental stimuli. (**C**) Screen composition and sequence in the emotional influence assessment task.

**Figure 2 brainsci-14-00637-f002:**
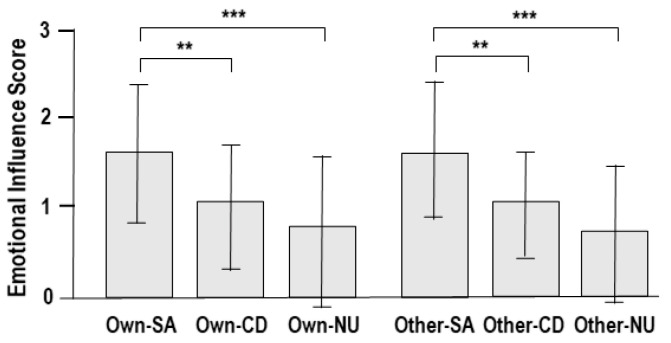
Behavioral responses in four experimental and two control conditions: own-voice and self-affirmation (Own-SA), own-voice and cognitive defusion (Own-CD), own-voice and neutral (Own-NU), other-voice and self-affirmation (Other-SA), other-voice and cognitive defusion (Other-CD), and other-voice and neutral (Other-NU) conditions. ** *p* < 0.01, *** *p* < 0.001.

**Figure 3 brainsci-14-00637-f003:**
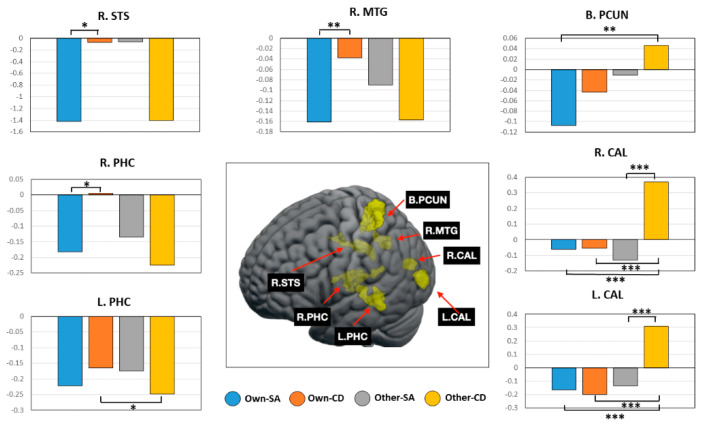
Brain regions showing the interaction effect between voice identity and emotion regulation strategy and comparisons of regional activity according to the four conditions of own or other-voice and self-affirmation (SA) or cognitive defusion (CD). * *p* < 0.05, ** *p* < 0.01, *** *p* < 0.001.

**Figure 4 brainsci-14-00637-f004:**
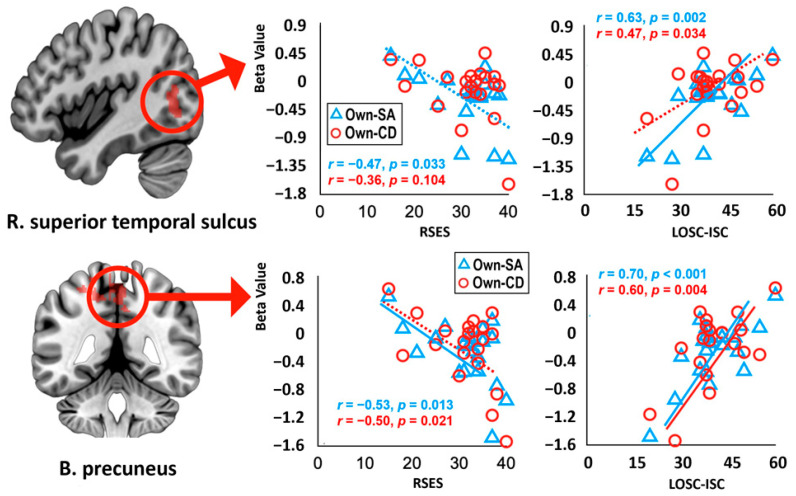
Significant correlations between regional activity and psychological assessments in the brain regions showing the interaction effect between voice identity and emotion regulation strategy. The solid and dotted lines in the graph represent significant and non-significant results after Bonferroni correction, respectively. Abbreviation: B., bilateral; R., right; Own-CD, the own-voice cognitive defusion condition; Own-SA, the own-voice self-affirmation condition; RSES, Rosenberg Self-Esteem Scale; LOSC-ISC, Levels of Self-Criticism Scale–Internalized Self-Criticism.

**Table 1 brainsci-14-00637-t001:** Brain regions showing increased activity during experimental conditions compared to the neutral condition.

Effect	Region	MNI Coordinate (x y z)			Nvox	Zmax
Own-SA	None					
Own-CD	L. Precentral gyrus	−44	−4	34	249	4.77
	L. Brainstem	−6	−32	−14	146	4.56
	L. Superior parietal lobule	−28	−46	42	187	4.55
	R. Inferior frontal gyrus	38	22	8	388	5.50
	L. Fusiform gyrus	−42	−42	−20	189	4.13
Other-SA	None					
Other-CD	L. Lingual gyrus	−14	−92	−10	346	4.37

Abbreviation: MNI, Montreal Neurological Institute; Nvox, number of voxels; Zmax, Z score of the peak voxel; L., left; R., right; SA, self-affirmation; CD, cognitive defusion.

**Table 2 brainsci-14-00637-t002:** Brain regions showing the main and interaction effects of voice identity and emotion regulation strategies.

Effect	Region	MNI Coordinate (x y z)			Nvox	Zmax	Direction
Identity	None						
Strategy	B. Medial prefrontal cortex	0	60	18	763	4.85	SA > CD
	L. Medial prefrontal cortex	−22	28	44	311	4.58	SA > CD
	L. Premotor cortex	−44	−2	34	669	5.76	CD > SA
	L. Superior temporal gyrus	−56	4	−6	164	4.55	SA > CD
	L. Inferior temporal gyrus	−42	−62	−14	1750	6.43	CD > SA
	R. Fusiform gyrus	36	−78	−14	688	5.19	CD > SA
Identity × Strategy	R. Superior temporal sulcus	44	−54	8	414	4.94	
	R. Middle temporal gyrus	32	−72	10	160	3.86	
	L. Parahippocampal cortex	−32	−42	−10	414	4.58	
	R. Parahippocampal cortex	18	−32	−14	672	5.18	
	B. Precuneus	−18	−44	52	651	4.26	
	L. Calcarine cortex	−14	−90	−4	223	4.77	
	R. Calcarine cortex	14	−92	−2	134	4.76	

Abbreviation: MNI, Montreal Neurological Institute; Nvox, number of voxels; Zmax, Z score of the peak voxel; L., left; R., right; B., bilateral; SA, self-affirmation; CD, cognitive defusion.

## Data Availability

The data are available from the corresponding author upon reasonable request due to patient privacy protection purposes.

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
