# Peer review of "Neural Effects of One’s Own Voice on Self-Talk for Emotion Regulation"

_brainsci, 2024, doi:10.3390/brainsci14070637_

Round 1

Reviewer 1 Report

Comments and Suggestions for Authors

Dear colleagues!

Thank you for the opportunity to review relevant research performed at a high level.

I have a few questions.

1. How did you calculate the sample size?

2. It will be useful in the discussion to develop the topic of epidemiology of the pathological process and the connection with your results

3. The list of references contains articles that are more than 20 years old. Should be updated

Author Response

Thank you for the opportunity to review relevant research performed at a high level.

Response:

We appreciate your positive evaluation of our research.

  1. How did you calculate the sample size?

Response: The sample size was calculated through a power analysis (G*Power 3, Faul et al., 2007) with a medium effect of 0.5 and 80% power at an alpha level of 0.05. Accordingly, 27 participants were originally recruited, but 6 participants were eliminated during the preparation of the experimental stimuli, and 21 participants ultimately participated in the experiment. In the original manuscript, only these 21 participants were described, but in the revised manuscript, the initial number of participants, the method of calculating the sample size, and the reason for elimination were added as follows.

Revisions:

2.1. Participants:

A total of 27 healthy right-handed adults (12 males / 15 females) were initially recruited into the study. Sample size was calculated through a power analysis (G*Power 3) [49] with a medium-sized effect of 0.5 and 80% power at an alpha level of 0.05. Exclusion criteria included the presence of a neurological, psychiatric, or significant medical illness, current or past history of substance abuse or dependence, and participation in a psychotherapeutic setting. Due to technical difficulties in synthesizing the own voice during the process of generating audio-visual stimuli for fMRI scanning, the stimuli for six participants did not reach a level suitable for the experiment. Accordingly, the remaining 21 participants (8 males / 13 females, mean age ± standard deviation: 25.6 ± 4.3) for whom auditory-visual stimuli were successfully prepared, participated in the fMRI experiment.

  1. It will be useful in the discussion to develop the topic of epidemiology of the pathological process and the connection with your results.

Response: As suggested, we linked our results to the clinical significance of using one's own voice in pathological processes and added the following to the discussion.

Revisions:

4th Paragraph of Discussion:

Previous literature has revealed that emotion regulation strategies, including mindfulness, are effective in controlling pathological factors, such as anxiety, depression, and stress, that underlie numerous clinical diseases [66,67]. Our findings may provide a brain basis for the effectiveness of using one's own voice to implement such strategies.

  1. The list of references contains articles that are more than 20 years old. Should be updated.

Response: Among the 61 references in the original manuscript, six were more than 20 years old. Among these six, the literature related to the self-esteem measurement scale was left alone as it was not appropriate to replace it with the latest one, and the remaining five were replaced with the latest literature as suggested.

Revisions:

References:

  1. Dolcos, S.; Albarracin, D. The inner speech of behavioral regulation: intentions and task performance strengthen when you talk to yourself as a you. Eur. J. Soc. Psychol. 2015, 44, 636e642.
  2. Dang, Q.; Wu, J.; Bai, R.; Zhang, B. Self-affirmation training can relieve negative emotions by improving self-integrity among older adults. Curr. Psychol. 2023, 42, 8816-8823.
  3. Cancino-Montecinos, S.; Björklund, F.; Lindholm, T. A general model of dissonance reduction: Unifying past accounts via an emotion regulation perspective. Front. Psychol. 2020, 11, 540081.
  4. Lindsaya, E.K.; Creswella, J.D. Mechanisms of mindfulness training: Monitor and acceptance Theory (MAT). Clin. Psychol. Rev. 2017, 51, 48–59.
  5. Goldberg, S.B.; Tucker, R.P.; Greene, P.A.; Davidson, R.J.; Wampold, B.E.; Kearney, D.J.; Simpson, T.L. Mindfulness-based interventions for psychiatric disorders: A systematic review and meta-analysis. Clin. Psychol. Rev. 2018, 59, 52-60.

Reviewer 2 Report

Comments and Suggestions for Authors

Neural Effects of One’s Own Voice on Self-talk for Emotion Regulation

I read the manuscript with interest and the authors can find my suggestions, appraisal, and commentaries, section-by-section, as follows:

Introduction: According to me, the introduction needs to be restructured. The interesting topic about the voice and the self-talk emotion regulation is introduced sparsely. This is not a bad criticism. The authors should introduce the “own voice-listening effect”. We like our voice until we do not speak publicly in a microphone that amplifies it and we think “It is not my voice, it is horrible, etc.” This is a complex process, mainly analyzed by cognitive psychology, and a better explanation is needed. Then you can introduce the cerebral underpinnings. Moreover, the authors introduced ex abrupto emotion regulation. The link is not obvious and maybe this needs to be rewritten in a better way.  The same is true for self-affirmation: the link is not clear and intelligible. Moreover, Self-talk: What is the difference between inner thoughts and self-affirmation talk?

Methods: participants. The authors need to add a wider description of the participants: past neuropsychiatric history, familiarization with psychotherapy settings, etc.

The RSES and LOSC (acronyms must be explained in the methods), need to be described in a better way. Similarly, Likert scale: specify, please, if they are Likert-like or Likert scales.

The procedure is interesting and well written, I recommend adding a schematic figure to facilitate the reading.

I invite the authors to add more information about the fMRI data analysis. In this way, I advise you to follow the COBIDAS checklist. Following the checklist, you can be sure to add all the relevant information, allowing replicability.

The results are interesting, but “flexible repeated-measures ANOVA” is not clear.

The fMRI results are also interesting. However, I advise you to discuss them with caution. Parahippocampus is also related to surprise, but, at same time is part of DMN and it is sensitive to the elaboration of naturalist content. The same for mPFC and precuneus.

These brain regions, part of DMN need to be discussed in a better way. DMN during the resting state works on the inner states, but during audiovisual stimulation, its activity (it is active during a naturalistic stimulation) is related to the specific content.

The authors can find here some insight: Yeshurun Y, Nguyen M, Hasson U. The default mode network: where the idiosyncratic self meets the shared social world. Nat Rev Neurosci. 2021 Mar;22(3):181-192. doi: 10.1038/s41583-020-00420-w. (not mine) and also take a look to : Brandman T, Malach R, Simony E. The surprising role of the default mode network in naturalistic perception. Commun Biol. 2021 Jan 19;4(1):79. doi: 10.1038/s42003-020-01602-z. (not mine) or simply read some papers from Uri Hasson. 

Author Response

Reviewer 2: 

I read the manuscript with interest and the authors can find my suggestions, appraisal, and commentaries, section-by-section, as follows:

  1. Introduction: According to me, the introduction needs to be restructured. The interesting topic about the voice and the self-talk emotion regulation is introduced sparsely. This is not a bad criticism. The authors should introduce the “own voice-listening effect”. We like our voice until we do not speak publicly in a microphone that amplifies it and we think “It is not my voice, it is horrible, etc.” This is a complex process, mainly analyzed by cognitive psychology, and a better explanation is needed. Then you can introduce the cerebral underpinnings. Moreover, the authors introduced ex abrupto emotion regulation. The link is not obvious and maybe this needs to be rewritten in a better way.  The same is true for self-affirmation: the link is not clear and intelligible. Moreover, Self-talk: What is the difference between inner thoughts and self-affirmation talk?

Response: We agree with the point that the introduction is not written systematically. To improve this, in the revised manuscript we changed the order of the descriptions to self-talk, voice effect, and emotion regulation. Additionally, as suggested, descriptions were added about the effects of listening to one's own voice, the inconvenience of listening to a recorded voice, the categories of emotion regulation, and the difference between self-talk and inner speech.

Revisions:

1st – 3rd paragraph of Introduction: 

Self-talk is the way people talk to themselves, their inner voice. A functional description of self-talk includes self-directed verbal expressions, encompasses various dimensions, such as positive or negative, overt or covert, and instructional or motivational, and involves elements of interpretation linked to the context of the statements used [1]. While inner speech is an unstructured stream of mental activity that includes voluntary or involuntary thought and reflection, self-talk is an attempt at self-regulation that occurs in response to or anticipation of a specific event or situation [2]. In particular, positive self-talk for self-affirming may play a crucial role in influencing decision-making [3], facilitating emotion regulation [4], and adapting to challenges [5], and thus has been employed in a variety of activities, such as enhancing performance in sports [6,7], boosting academic involvement [8], and managing anxiety in public speaking [9].

Self-talk inevitably leads to hearing one's own voice. Individuals process their own voices differently from others' voices in ways that perceive them as more attractive [10,11]. This attractiveness can be explained by vocal implicit egoism, a form of self-enhancement driven by the familiarity effect and self-positivity bias [12]. Phonetic realizations of one's own voice significantly shape phonological contrasts, leading to more accurate recognition of words in one's own voice compared to the voices of others [13]. In addition, as individuals become accustomed to their own voices through lifelong exposure, hearing their own voice exhibits the phenomenon of neural sharpening, in which more common stimuli reduce neural responses to them, and thus it lowers the level of activation of the superior temporal gyrus (STG), which is involved in neural sharpening for voices [14]. Furthermore, a previous neuroimaging study have shown that hearing the own voice causes engagement of the self-referential network, including the medial prefrontal and parietal cortices [15], supporting that it is linked to self-awareness in speech processing. Since people listen to their own voices while speaking, they perceive their voices more deeply and richly through bone conduction and air conduction. Sometimes people listen to their own recorded voices, which they hear only through air conduction, making them feel uncomfortable because they are different from their familiar voices [16]. Therefore, if an experimental attempt is planned to investigate the effect of self-talk, it is preferable to listen to one's own recorded voice rather than someone else’s voice, and the process of converting this recorded voice into sound like the voice heard when speaking is first required. This kind of investigation may be possible by measuring electrodermal activity, a physiological signal for objective assessment of emotional states [17], or functional MRI, a powerful technique that captures brain responses to task-related activities with high spatial resolution [18].

Self-talk may be one of emotion regulation strategies. Emotion regulation is a process of controlling one’s own emotional state. A variety of emotion regulation strategies have been developed to improve mental health in several different categories, such as attention allocation, response regulation, reappraisal, and suppression [19,20]. These strategies have been reported to be associated with multiple cortical and subcortical activations in the brain [21,22]. Self-talk for self-affirming may be an ex-ample of practical regulatory attempts.

  1. Methods: participants. The authors need to add a wider description of the participants: past neuropsychiatric history, familiarization with psychotherapy settings, etc.

Response: Because the participants were all healthy volunteers, the exclusion criteria were described in the existing description. As suggested, participation in a psychotherapeutic setting was also added in the exclusion criteria in the revised manuscript.

Revisions:

2.1. Participants: 

Exclusion criteria included the presence of a neurological, psychiatric, or significant medical illness, current or past history of substance abuse or dependence, and participation in a psychotherapeutic setting.

  1. The RSES and LOSC (acronyms must be explained in the methods), need to be described in a better way. Similarly, Likert scale: specify, please, if they are Likert-like or Likert scales.

Response: As suggested, the full names of the two scales were specified in the methods section, and the specific details of the Likert scale and total scores were added.

Revisions:

2.2. Psychological Assessments: 

They were the Rosenberg Self-Esteem Scale (RSES), consisting of 10 items with 4-point Likert scale from 1 (strongly disagree) to 4 (strongly agree) and a total score range from 10 to 40 [50], and the Levels of Self-Criticism Scale (LOSC), consisting of a 22-item 7-point Likert scale anchored by 1 (not at all) and 7 (very well) [51]. The LOSC includes two subscales: comparative self-criticism (12 items, total score range: 12-82) and internalized self-criticism (10 items, total score range: 10-70).

  1. The procedure is interesting and well written, I recommend adding a schematic figure to facilitate the reading.

Response: As suggested, a new schematic figure of the experimental procedure was created and inserted into Figure 1B. Instead, the graph of behavioral outcome in previous Figure 1C was converted to Figure 2.

Revisions:

Figure 1. The experiment overview: (A) Four experimental conditions, such as the own voice and self-affirmation, the own voice and cognitive defusion, the other’s voice and self-affirmation, and the other’s voice and cognitive defusion, (B) schematic diagram of the experimental procedure, including participant visitation and preparation of experimental stimuli, and (C) screen composition and sequence in the emotional influence assessment task.

Figure 2. Behavioral responses in four experimental and two control conditions: the own voice and self-affirmation (Own-SA), the own voice and cognitive defusion (Own-CD), the own voice and neutral (Own-NU), the other’s voice and self-affirmation (Other-SA), the other’s voice and cognitive defusion (Other-CD), and the other’s voice and neutral (Other-NU). **p < 0.01, ***p < 0.001

  1. I invite the authors to add more information about the fMRI data analysis. In this way, I advise you to follow the COBIDAS checklist. Following the checklist, you can be sure to add all the relevant information, allowing replicability.

Response: Thank you for your valuable comment. As suggested, we referred to the COBIDAS checklist and tried to add as much image analysis content as possible without departing from the context.

Revisions:

2.6. Imaging Data Analysis: 

Preprocessed functional data were analyzed using a general linear model at the single-subject level. Experimental trials were modeled separately using a canonical hemodynamic response function for individual data. Multiple linear regression was used to obtain parameter estimates using a least-squares approach. These estimates were further analyzed by testing specific contrasts using the participant as a random factor. Contrast images of four experimental conditions subtracted by the neutral control condition were created for each participant on the first-level analysis. Individual realignment parameters were entered as regressors to control for movement-related variance. In order to find brain activations in each experimental condition, the contrast images were entered into the one-sample t-test and the full factorial model across the participants. In addition, in order to find common activation areas of the two emotion regulation strategies, a conjunction analysis was performed between contrast images of the self-affirmation and cognitive defusion conditions.

  1. The results are interesting, but “flexible repeated-measures ANOVA” is not clear.

Response: Thank you for pointing out our mistake. “Flexible” wasn’t a word that needed to be included. So we deleted it from the revised manuscript.

  1. The fMRI results are also interesting. However, I advise you to discuss them with caution. Parahippocampus is also related to surprise, but, at same time is part of DMN and it is sensitive to the elaboration of naturalist content. The same for mPFC and precuneus.

These brain regions, part of DMN need to be discussed in a better way. DMN during the resting state works on the inner states, but during audiovisual stimulation, its activity (it is active during a naturalistic stimulation) is related to the specific content.

The authors can find here some insight: Yeshurun Y, Nguyen M, Hasson U. The default mode network: where the idiosyncratic self meets the shared social world. Nat Rev Neurosci. 2021 Mar;22(3):181-192. doi: 10.1038/s41583-020-00420-w. (not mine) and also take a look to : Brandman T, Malach R, Simony E. The surprising role of the default mode network in naturalistic perception. Commun Biol. 2021 Jan 19;4(1):79. doi: 10.1038/s42003-020-01602-z. (not mine) or simply read some papers from Uri Hasson.

Response: We are very grateful to you for suggesting important implications about the role of the DMN. This has significant implications for interpreting our results. In the revised manuscript, we cited the two documents you suggested and included the following important concepts in the discussion.

Revisions:

3rd Paragraph of Discussion: 

The precuneus, along with the MPFC, is involved in self-referential processing as part of the default mode network [61,62]. This network is responsible for integrating moment-to-moment external information with prior information, and thus its activity is influenced by context and incoming input [63].

4th Paragraph of Discussion: 

A recent study reported that the default mode network including the precuneus may be involved in external naturalistic event processing and prediction-based learning [65], suggesting that this network can be changed by applying an appropriate learning strategy.

Reviewer 3 Report

Comments and Suggestions for Authors

1. The objective of the study is not clear.

2. Write separate Novelty (in a paragraph) and contribution (in bullet points) after the introduction.

3. What is the reason behind choosing fMRI over other methods?

4. The physiological methods EDA and EEG have shown significance accurately in emotion-related studies. Why did the authors consider fMRI over these methods?

5. The literature included in the manuscript needs to be improved. Including the current state-of-the-art works in emotion recognition such as: https://doi.org/10.1109/ACCESS.2024.3361832; https://doi.org/10.1109/JSEN.2024.3354553 will improve the readability of the paper.

6. Are stimuli induced randomly to the participants?

7. The reason behind choosing two-way ANOVA over other methods is not clear. Elaborate.

8. The results section starts with fMRI representative images in 4 experimental conditions and indicates the differences.

9. Can you extract features and classify the 4 experimental conditions?

10. Write limitations and future scope of the study in the discussion section.

Comments on the Quality of English Language

NA

Author Response

Reviewer 3

 1. The objective of the study is not clear.

Response: In order to make the objective of the study clearer, the related sentences were rewritten as follows.

 Revisions:

The last paragraph of Introduction

The current study used a task of listening to sentences of the emotion regulation strategies and assessing their emotional influence while undergoing functional MRI. Given the uniqueness of one's own voice, it may be more efficient to perform this task by listening to one's own voice rather than listening to the voices of others, and this efficiency may vary in degree depending on the type of emotion regulation strategy. The purpose of the current study was to elucidate how the neural effects of one’s own voice differ from those of others’ voices on implementing the voice-listening emotion regulation strategies, such as self-affirmation and cognitive defusion.

 2. Write separate Novelty (in a paragraph) and contribution (in bullet points) after the introduction.

Response: As suggested, we added the following to the introduction to highlight novelty and contribution.

 Revisions:

The last paragraph of Introduction

To our knowledge, an fMRI study like this has not been conducted before, and will contribute to providing a foundation and understanding of the importance of using one's own voice in the development of emotion regulation strategies.

3. What is the reason behind choosing fMRI over other methods?

Response: A number of pros and cons of functional MRI compared to other imaging methods could be listed, but these are far from the focus of this study, so there is no need to describe them in detail in our paper. However, to emphasize that it is reasonable to use functional MRI for the purpose of our study, we briefly added one key advantage to the description of the revised manuscript as follows.

 Revisions:

2nd paragraph of Introduction

This kind of investigation may be possible by measuring electrodermal activity, a physiological signal for objective assessment of emotional states [17], or functional MRI, a powerful technique that captures brain responses to task-related activities with high spatial resolution [18].

4. The physiological methods EDA and EEG have shown significance accurately in emotion-related studies. Why did the authors consider fMRI over these methods?

Response: As pointed out, it is clear that EDA and EEG are excellent means for studying emotions. However, as mentioned in the previous #3 answer, in this study to find neural substrates, functional MRI with high spatial resolution was used, and a brief description about this was added to the revised manuscript.

 5. The literature included in the manuscript needs to be improved. Including the current state-of-the-art works in emotion recognition such as: https://doi.org/10.1109/ACCESS.2024.3361832; https://doi.org/10.1109/JSEN.2024.3354553 will improve the readability of the paper.

Response: Thank you for pointing out the excellent literature published recently. We quoted one of these for context as follows.

Revisions:

2nd paragraph of Introduction

This kind of investigation may be possible by measuring electrodermal activity, a physiological signal for objective assessment of emotional states [17], or functional MRI, a powerful technique that captures brain responses to task-related activities with high spatial resolution [18].

17. Veeranki, Y.R.; Diaz, L.R.M.; Swaminathan, R.; Posada-Quintero, H.F. Nonlinear signal processing methods for automatic emotion recognition using electrodermal activity. IEEE Sens. J. 2024, 24, 8079-8093

 6. Are stimuli induced randomly to the participants?

Response: Two sets of the randomly arranged stimuli were produced and presented alternately to the participants. This was added in the description of study design as follows.

 Revisions:

2.3. Audiovisual Stimuli and Experimental Procedure

In the sequence of the fMRI experiment, 120 pre-generated audio-visual stimuli were randomly placed, and each stimulus was presented for seven seconds at jittered intervals of one to seven seconds (Figure 1C). Two sets of these randomly arranged stimuli were produced and presented alternately to the participants.

7. The reason behind choosing two-way ANOVA over other methods is not clear. Elaborate.

Response: As suggested, the reason was clarified in the revised manuscript as follows.

 Revisions:

2.5. Behavioral Response Analysis

In order to assess the main effect of each of two categorical independent variables, voice identity and emotion regulation strategy, and the interaction effect between them, the emotional influence scores were compared in a 2 (identity: own and the other) × 3 (strategy: self-affirmation, cognitive defusion, and neutral) manner using two-way analysis of variance (ANOVA).

8. The results section starts with fMRI representative images in 4 experimental conditions and indicates the differences.

Response: As suggested, we presented the results of the first-level analysis at the beginning of the imaging results in the revised manuscript. These results were presented in a table, but since they were not the core of our study, we did not present them in a figure.

 9. Can you extract features and classify the 4 experimental conditions?

Response: The answer to this is the same to the answer to # 8.

 10. Write limitations and future scope of the study in the discussion section. 

Response: Since the limitations were described in a separate paragraph, future scope of the study was added to the conclusion as follows.

 Revisions:

5. Conclusions

These insights into brain responses may be important for developing personalized treatment approaches to improve mental health, taking into account individual differences and preferences. From this perspective, future research will be needed to determine how one's own voice affects the brain in emotional regulation strategies other than self-affirmation and cognitive defusion.

Round 2

Reviewer 2 Report

Comments and Suggestions for Authors

According to me, the manuscript has been improved.

The authors addressed all the issues that l have raised. However, the manuscript needs to check for typos. Please, check the English language 

Reviewer 3 Report

Comments and Suggestions for Authors

Authors have incorporated all my suggestions